# The Scattering and Neutrino Detector at the LHC

F.L. Navarria[1,2] on behalf of the SND@LHC Collaboration

**1** University of Bologna, I-40127 Bologna, Italy
**2** INFN, Sezione di Bologna, I-40127 Bologna, Italy
* francesco.navarria@bo.infn.it

December 19, 2021

## Abstract

**SND@LHC is a compact experiment that will detect high energy neutrinos produced by heavy flavour quarks at the LHC in the pseudo-rapidity region $7.2 < \eta < 8.6$. It is an hybrid system, comprising nuclear emulsions and electronic detectors, that allows all three $\nu$ flavours to be distinguished, thus opening an unique opportunity to probe the physics of charm production in the very forward region. The first phase aims at operating the detector throughout LHC Run 3 collecting a total of 150 fb$^{-1}$. The electronic subdetectors were assembled in the summer and recently operated in test beams.**

## 1 Introduction

As the highest energy particle accelerator built so far, the Large Hadron Collider (LHC) is the source of the most energetic neutrinos ($100-3000$ GeV) created in a controlled laboratory environment. The detection of the abundant $\nu_e$, $\nu_\mu$, $\nu_\tau$ flux from the decay of W bosons and heavy flavour quarks in proton-proton (pp) collisions at the LHC can fill the gap between accelerator measurements [1] (mostly with $\nu_\mu$, few with $\nu_e$, only 19 $\nu_\tau$ candidates seen [2] [3]) and observations with cosmic rays [4]. During the LHC Run 3, $2022-2025$, with the foreseen integrated luminosity of $\sim$150 fb$^{-1}$, an $O$(ton) detector with good tracking and particle identification capabilities can unveil a sizeable number of $\nu$ interactions. Neutrinos from W boson decays would a priori be more interesting as they are a democratic mix of all $\nu$ and $\bar{\nu}$ flavours. The detection of the corresponding charged leptons in ATLAS/CMS could provide an unique opportunity of tagging $\nu$ and $\bar{\nu}$ flavour with good efficiency [5]. However, tests at different distances from an LHC interaction point (IP5) exclude this possibility. A pseudorapidity, $\eta \sim$ 4.5, ideal for $\nu$s from W boson decays, implies a distance from IP $\sim$ 25 m in order to keep the detector small: in this region we encountered an overwhelming neutron background (n fluence $\sim 3x10^9$/cm$^2$/fb$^{-1}$ [6] [7]). The other possibility is to detect $\nu$ from c, b quarks decay, and $\eta \sim 8$, $\theta \sim 0.7$ mrad at a distance 480 m from IP1 was chosen for a compact detector located in the unused TI18 [8]. TI18, a former service tunnel connecting SPS to LEP, is symmetric w.r.t. TI12 where FASER$\nu$ [9] is located. The line of sight to IP1 is shielded by $\sim$ 100 m rock, so only $\mu$ and thermal n from beam-gas interaction (and $\nu$s!) survive. Charged particles coming from IP are also swept away by LHC magnets. SND@LHC (SND for short in the following) was approved by the CERN Research Board in March 2021.

Together with FASER$\nu$, SND will observe the first high energy $\nu$s produced by a collider. Two distinct detector concepts, SND and FASER$\nu$, will explore different angular ranges in

which the relative compositions of the various sources of neutrinos are different. Contrary to FASER$\nu$, which is on-axis, $\eta > 9$, the SND detector is offset w.r.t. the collision axis, motivated both by the physics [6] [7], as very forward $\nu$ are dominated by $\pi$ and K decays, and by avoiding excavation in the TI18 tunnel. Thus the SND angular acceptance is $7.2 < \eta < 8.6$, and the neutrino physics programmes of the two experiments are complementary.

With data from Run 3, SND will be able to study at least fifteen hundred high-energy $\nu$ interactions, and to detect about 20 $\nu_\tau$. Performance simulations of collider measurements show that the charmed-hadron production in the SND pseudorapidity range can be determined with an overall uncertainty of 35%. Fixed target measurements produce unique tests of lepton flavour universality with neutrino interactions that can reach an overall uncertainty of 35% for $\nu_e$ vs $\nu_\tau$, and 15% for $\nu_e$ vs $\nu_\mu$ at high energy.

## 2   The detector

The detector (Fig. 1) is a compact hybrid system, containing passive and active subdetectors. It comprises a target section consisting of emulsion cloud chambers (ECCs) interleaved with a scintillator fiber tracker (SciFi), and a muon detector/hadron calorimeter with a final muon tracking section [8].

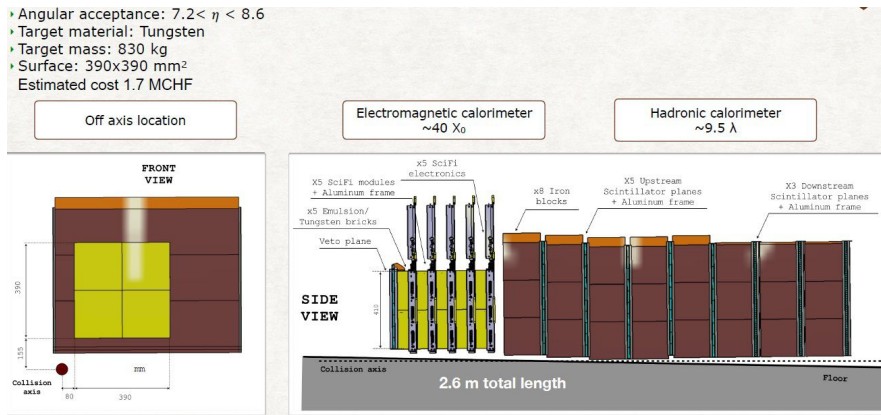

Figure 1: SND detector

The target region is protected by a veto wall made of plastic scintillators that detects impinging charged particles. It is made of two vertically shifted planes of seven 42x6x1 cm$^3$ bars, read at each end by SiPMs placed on a PCB common to each side of a detector plane. Each ECC comprises 60 nuclear emulsion films, 19.2x19.2 cm$^2$, with 59 1 mm thick W-alloy plates in between, for a weight of 41.5 kg. The total weight of the target is about 830 kg. The total emulsion surface is about 44 m$^2$. SciFi consists of five 40x40 cm$^2$ $x$-$y$ planes of staggered scintillating fibers each 250 $\mu$m in diameter read by customized 128 channel Hamamatsu SiPMs. Apart from tracking, SciFi plays also the role of a coarse sampling (16.8 X$_0$ or 0.593 $\lambda_I$) e.m./hadron calorimeter. For single hits/tracks the space resolution is $\sim$50 $\mu$m, sufficient to link the hits with an interaction in an EEC brick, and the time resolution is $\sim$ 250 ps. For multiple tracks or showers, the time resolution is expected to be considerably better. For the long-term stability of emulsion films, the temperature of the target will be kept at 15 +- 1 °C and the relative humidity in the range 50 to 55%. For this purpose an insulated box is built around the target region and a cooling system is installed. The walls of the box act as a passive neutron shield with a borated polyethylene layer.

The muon system consists of two parts, upstream (US), first five stations, and downstream (DS), last three stations. It plays, in combination with SciFi, the role of a hadron calorimeter,

catching the tail of the hadronic shower leaking out of the target region. The sampling in the $\mu$ system is also coarse, every 20 cm of Fe or 11.4 $X_0$ or 1.19 $\lambda_I$, so US acts also as a $\mu$ filter, while DS is used to track and identify the muon. On average, an electron produced in the centre of the target will see 42 $X_0$ in the target region, followed by 91 $X_0$ in the muon system, while a hadron will see 1.5 $\lambda_I$ in the target region, followed by 9.5 $\lambda_I$ in the muon system. Each US station consists of 10 stacked horizontal scintillator bars, (82.5x6x1) $cm^3$, which are read out at the opposite ends by SiPMs. Each DS station comprises 60 horizontal bars, (82.5x1x1) $cm^3$, and 60 vertical ones, (63.5x1x1) $cm^3$, allowing for a space resolution of better than 1 cm in the $x$-$y$ directions. The horizontal bars are read out on each side by SiPMs, and the vertical bars are read out only from the top. A fourth vertical layer is added to the last station to allow for some redundancy given the readout from only one end. In all stations, the SiPMs are followed by front-end electronics and TOFPET2 ASICs. The planes of scintillator bars have a timing resolution better than 100 ps for time-of-flight measurements of particles from the ATLAS IP. The resolution in time-of-flight will thus be determined by the 200 ps temporal spread that corresponds to the length of the luminous region at IP1.

The event reconstruction is performed in two stages. The first stage uses electronic detectors (veto, SciFi and muon system). The veto tags incoming $\mu$s that will be used for the alignment between detector planes (and for the fine alignment of the emulsion films). The occurrence of a $\nu$ interaction is first detected by the target tracker and the muon system. Electromagnetic showers are expected to be absorbed within the target region and are therefore identified by the target tracker, while $\mu$s in the final state are reconstructed by the muon system. Using the DS stations, the overall $\mu$ identification (ID) efficiency is $\sim$ 69% in simulated charged current (CC) $\nu_\mu$ interactions, with a purity of almost 99%. In addition, the detector as a whole acts as a sampling calorimeter. The combination of data taken from both SciFi and $\mu$ systems is used to measure the hadronic and electromagnetic energy of the event. The average energy resolution for showers in simulated CC $\nu_e$ interactions is found to be $\sim$ 22% using simple counting, $E_h^{rec} = A + B\ N_{SciFi} + C\ N_{\mu filter}$. Machine Learning algorithms are being developed to improve the resolution.

The second stage uses nuclear emulsions (event reconstruction in the emulsion target) and provides: identification of e.m. showers; primary vertex reconstruction and secondary vertex search; matching with candidates from electronic detectors (acquiring a time stamp). Without going into details of the event recostruction in the EECs [8], we show in Fig. 2 the topology of some signal events that can be reconstructed in a brick. The identification of the neutrino flavour is done in CC interactions by identifying the charged lepton produced at the primary vertex. Electrons are clearly separated from $\pi^0$s thanks to the micrometric accuracy of the EEC, which enables photon conversions downstream of the neutrino interaction vertex to be identified. The electron ID efficiency in the interaction brick is $>$ 95%, and reaches $\sim$ 99% including the SciFi information. Muon identification at vertex in emulsions is crucial to identify charmed hadron production, which is a background to $\tau$ detection. Tau leptons are identifiied topologically in the emulsion, through the observation of the tau decay vertex, together with the absence of any electron or muon at the primary vertex. The $\tau$ decay search (total ID) efficiency ranges from 80−82% (48−50%) for 1-prong events to 89% (54%) for 3-prongs.

# 3  Physics goals

Neutrino production in pp collisions at the LHC is simulated with FLUKA [10], with DPMJET3 (Dual Parton Model, including charm) [11] used for event generation. FLUKA performs the particle propagation towards the SND detector with the help of a model of the LHC machine. It also takes care of simulating the production of neutrinos from decays of long-lived products

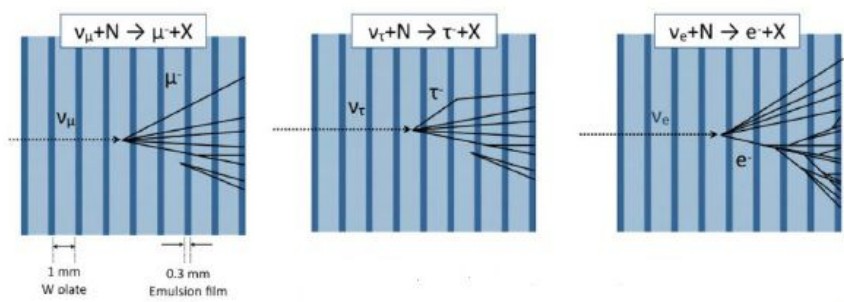

Figure 2: Neutrino CC interaction topologies

of the pp collisions and of particles produced in re-interactions with the surrounding material. GENIE [12] is then used to simulate neutrino interactions with the SND detector material. The output of GENIE is fed to GEANT4 [13] for particle propagation in the detector.

Neutrino fluxes vs $\eta_\nu$ and $E_\nu$ within the SND acceptance are shown in Fig. 3. The momentum of the two $\nu_\tau$ produced in the $D_s \rightarrow \tau\,\nu_\tau \rightarrow X\,\bar{\nu}_\tau\,\nu_\tau$ decay chain, causes a correlation between $\eta_\nu$ and $E_\nu$, clearly visible in the right panel. $\nu_e\,and\,\nu_\tau$ are mainly coming from the decay of charmed hadrons. 10% of $\nu_e$ interacting within the acceptance come from K decay, in particular $K_s^0$, and have energy below 200 GeV. The contribution of beauty-hadron decays at IP1 was estimated with the help of the PYTHIA8 [14] event generator to be about 3%. On the other hand, $\nu_\mu$ and $\bar{\nu}_\mu$ spectra are heavily affected by a soft component from $\pi$ and K decays. The energy spectra of neutrinos interacting within the SND acceptance are shown in Fig. 4 for 150 fb$^{-1}$.

The set of measurements feasible with neutrinos with 150fb$^{-1}$ at 13 TeV is summarized in Table 1. If the charged lepton is not identified, $\nu_\mu$ and $\nu_e$ CC events with charm production, about 10% of the total, are the main background in the $\nu_\tau$ search. With simple kinematics cuts, this background is substantially reduced [3], and the final signal-to-background ratio is $\sim 4$, as reported in the Table.

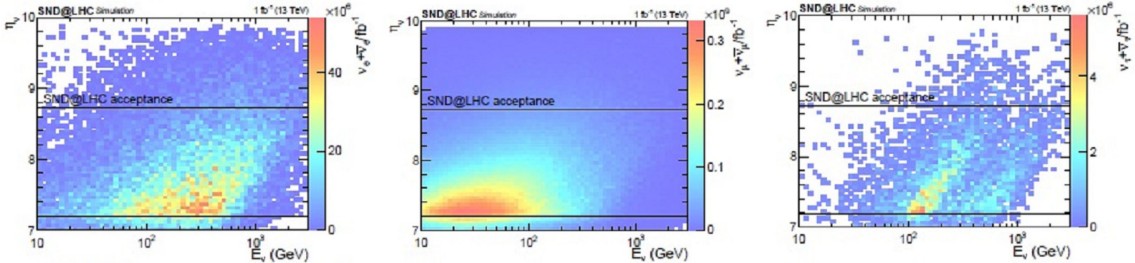

Figure 3: Neutrino fluxes vs $E_\nu$ and $\eta_\nu$ at 13 TeV for 1 fb$^{-1}$: left) $\nu_e$; middle) $\nu_\mu$; right) $\nu_\tau$.

## 3.1   Charmed-hadron production at large $\eta$ in pp collisions

If one assumes that the deep-inelastic scattering (DIS) CC cross section of the electron neutrino follows the Standard Model prediction, as supported by the HERA results [15], $\nu_e$s can be used as a probe of the production of charm in the pseudorapidity range of SND, after unfolding the instrumental effects and subtracting the K contribution. A procedure was developed that starts from the observed $\nu_e$ energy spectrum, and unfolds the energy resolution effects to predict the energy spectrum of the incoming $\nu_e$ flux. At this stage, data can be used to measure the pp $\rightarrow \nu_e$X cross section with an accuracy of 15%, dominated by the systematic uncertainty in the

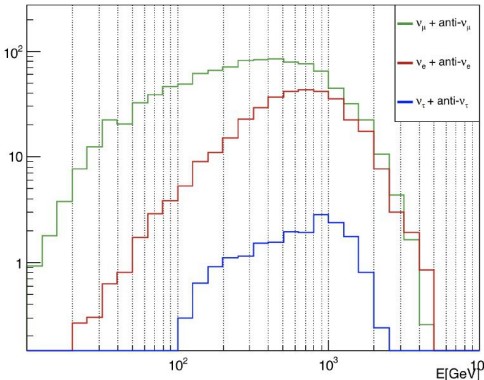

Figure 4: Energy spectra of interacting neutrinos for 150 fb$^{-1}$ at 13 TeV.

Table 1: Measurements proposed by SND in the analyses of neutrino interactions with Run 3 data. Estimated statistical and systematic uncertainties are indicated.

| Measurement | Stat.(%) | Syst.(%) | Signal/Bkgd |
|---|---|---|---|
| pp $\rightarrow \nu_e$X cross section | 5 | 15 | |
| Charmed hadron yield | 5 | 35 | |
| $\nu_e/\nu_\tau$ ratio for LFU test | 30 | 22 | |
| $\nu_e/\nu_\mu$ ratio for LFU test | 10 | 10 | |
| NC/CC ratio | 5 | 10 | |
| Observation of high-energy $\nu_\tau$ | | | 4 |

unfolding procedure. Different event generators predict different levels of K contribution, but all agree that the events are restricted to energies below 200 GeV. When subtracting the K contribution, the uncertainty on K production gives an additional uncertainty of 20% in the heavy-quark production. A detailed procedure with a full simulation was setup to correlate the yield of charmed hadrons in a given $\eta$ region with the neutrinos in the measured $\eta$ region, yielding another 25% systematic uncertainty in the charmed-hadron yield. As a result, the measurement of the charmed-hadron production in pp collision can be done with a statistical uncertainty of about 5% while the leading contribution to the uncertainty is the systematic error of 35%. From there, one can also derive information on the gluon parton distribution function in an unexplored $x$ region, as the charmed quark is produced via gluon-gluon scattering, one with $x \sim 2 \ 10^{-1}$, the other with $x \sim 2 \ 10^{-6}$, on average.

## 3.2 LFU tests in neutrino interactions

Cross section ratios can provide lepton flavour universality (LFU) tests in neutrino interactions. Here we consider two ratios, between $\nu_e$ and $\nu_\tau$, and bewteen $\nu_e$ and $\nu_\mu$.

$\nu_\tau$s are mainly coming from D$_s \rightarrow \tau \nu_\tau$ with about 8% from beauty decays. $\nu_e$ come from D$^0$, D, D$_s$ and $\Lambda_c$ decays. The cross section ratio. R$_{13} = N_{\nu_e}/N_{\nu_\tau}$, only depends on charm hadronization and decay branching fractions. The systematic uncertainty in the fraction of D$_s$ is studied using different generators, PYTHIA8 [14], PYTHIA6 [16], HERWIG [17], and turns out to be 22%. Uncertainties due to charm quark production cancel out. Therefore R$_{13}$ is sensitive to the $\nu$-nucleon interaction cross section ratio and allows LFU to be tested in neutrino interactions with a dominant 30% statistical uncertainty due to the $\nu_\tau$ sample size.

The branching fractions in charmed hadron decays for the production of $\nu_e$ and $\nu_\mu$ are practically equal, so they cancel out in taking the ratio R$_{12} = N_{\nu_e}/N_{\nu_\mu}$. In this case however one has a large contamination of $\nu_\mu$ from $\pi$ and K decays, which is stable above 600 GeV within

a 15% accuracy, such that the ratio of cross sections can be expressed as $R_{12} = 1/(1 + \omega_{\pi K})$, where $\omega_{\pi/K}$ is the contamination.

## 3.3 Measurement of the NC/CC ratio

Charged lepton ID for three flavours allows CC to be distinguished from neutral current (NC) interactions, but, given the absence of magnetic field it is impossible to distinguish CC $\nu$ and $\bar{\nu}$ interactions as there is no charge ID. Flavours of NC interactions are in any case indistinguishable, i.e. impossible to disentangle in a mixed beam. Assuming that the fluxes of $\nu$ and $\bar{\nu}$ as a function of $E_\nu$ are equal, the NC/CC cross section ratio, $R^+ = \Sigma_i(\sigma_{NC}^{\nu_i} + \sigma_{NC}^{\bar{\nu_i}})/\Sigma_i(\sigma_{CC}^{\nu_i} + \sigma_{CC}^{\bar{\nu_i}})$, is equal to the ratio of the observed events in the corresponding channels. A simple expression can be derived for $R^+$ as a function of the Weinberg angle [18]:

$$R^+ = \frac{1}{2} - sin^2\theta_W + \frac{10}{9}sin^4\theta_W - \lambda(\frac{1}{2} - sin^2\theta_W)sin^2\theta_W. \tag{1}$$

The last term in eq. 1 is a correction for the non isoscalar target, with $\lambda = 0.04$ for tungsten. The uncertainties go as follows. Systematics: $\sim$10% $\nu/\bar{\nu}$ spectra asymmetry, $\mu$ identification, neutron induced events, CC/NC migration. Statistical: $\sim$5% due to the number of NC interactions. This measurement will be used as a control measurement.

## 4 Detector assembly and status

It was literaly a rush programme. The technical proposal was submitted in January 2021 [8]. SND was approved in March. From April to August the active detector was built: veto, one SciFi plane that was missing, US and DS muon stations. This implied procuring the scintillator bars, wrapping them with aluminized mylar to ensure optical separation and assemblying them into support frames; designing and procuring the PCBs with the SiPMs. At the same time the infrastructure was being prepared at the installation site, with the installation in TI18 foreseen to start later in November. As from September the commissioning and calibration of various parts of the detector in test beams started.

As an example, we will give a few details of the DownStream muon section (Fig. 5). Two out of three planes were assembled by the end of August. Given the tolerances of the scintillator bars, the assembly of DS planes has required very careful wrapping with aluminized mylar and some special tooling to maintain the alignement with the PCBs with SiPMs. Vertical bars, read only at the top end, are wrapped with an aluminized mylar reflector at the opposite end to maximize light collection and guarantee full efficiency. A trained person could wrap on average 13 bars per day. The construction of the three DS stations, including the fourth vertical plane, was terminated in September.

## 5 Test beam

The muon US and and part of the DS section, preceded by an iron block to simulate interactions in the middle of the target, were calibrated with a $\pi^-$ beam between 100 and 180 GeV in the H6 line of the SPS at the beginning of September. The complete muon section and SciFi (Fig. 6) were measured in the same beam line at the beginning of October with $\pi^-$ of energies between 180 and 300 GeV, and later moved to the H8 beam line. The data are being analyzed.

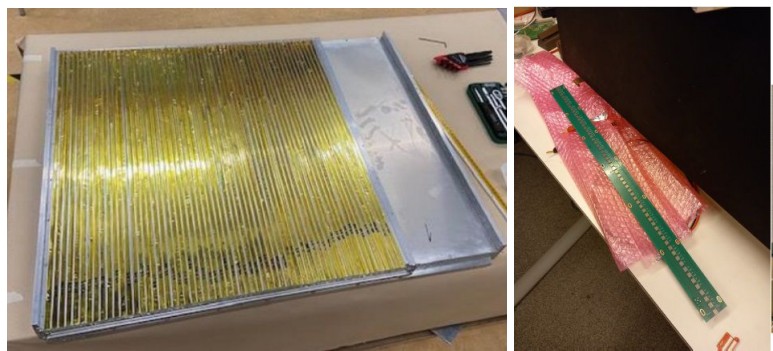

Figure 5: DS assembly: left) plastic scintillator bars wrapped with aluminized mylar and stacked into the supporting frame; right) the PCB with the SiPMs.

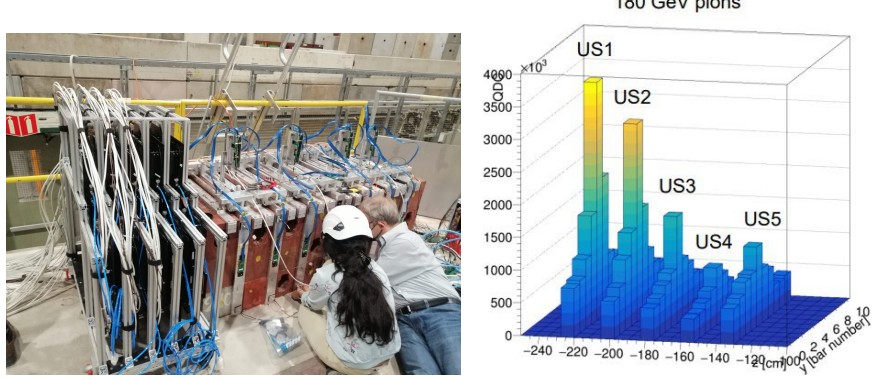

Figure 6: left) SciFi, US and DS muon section in the H6 test beam; right) preliminary hadronic shower profile in the US section, the lower charge yield in US4 is due to a poor TOPFET2 connection.

## 6 Conclusion

SND@LHC is capable of detecting the interactions of different neutrino flavours and should detect thousands of them in the next run of the LHC, starting in June 2022. The detector was assembled in the summer and calibrated in test beams. Installation in the TI18 tunnel started in November.

## Acknowledgements

I would like to thank my SND colleagues, and in particular M. Dallavalle and A. De Crescenzo for helping me with the preparation of the talk and of the writeup.

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
