# Peer review of "The Scattering and Neutrino Detector at the LHC"

_SciPost Physics Proceedings_

## Round 1 · Referee Report · Anonymous (Referee 1) · 2022-2-2

Report

This proceeding discusses the potential of and progress in instrumenting the SND compact experiment at forward rapidities at the LHC.

Overall, it is well written and provides a clear discussion on the physics goals and the recent progress in instrumenting the detector concept.

There are a few exceptions, where I believe the author could clarify some details before publication.

They are: 1. The presentation of Fig. 3 is a little unclear. For example, it is hard to understand what exactly what is the purpose of the figures. The y-axis of the plots have a different normalisation ($10^3$ different for muons) and the labels are hard to read. Is the point of the plot to show the relative contribution of the various flavours in the $\eta$ vs $E_{\nu}$ plane? In which case that could be mentioned, and perhaps a note on the different y-axis ranges could be highlighted. In the end, the translation of this information in Fig 4 to event rates is what is more important? 2. What is the y-axis in Fig. 4? Expected event rates after accounting for detection efficiencies? 3. In Table 1, a systematic uncertainty on the charm yield is quoted as $35%$. From the point of view of a theory prediction, that uncertainty is far far below that of the uncertainty on the cross-section prediction (see for example the comparison of the absolute D-hadron rates in measurement w.r.t. to theory in https://arxiv.org/pdf/1510.01707.pdf [13 TeV measurement by LHCb]). If instead, the charm yield will be taken from a combination of measurements at central (ALICE) and forward (LHCb) rapidities and then propagated to the far-forward region relevant for the future SND measurements, then that should mentioned. That is an important point, as otherwise the systematic uncertainty on the charm yield will be overwhelmingly large. 4. It should be noted that formula (1) is just an approximation. The hadronic cross-sections (and the ratio between CC and NC) depend in a complicated way on the flavour composition of the nuclear PDFs (and in addition the extrapolation to the low-momentum transfer regime where collinear factorisation is not applicable).

  • validity: good
  • significance: good
  • originality: good
  • clarity: high
  • formatting: good
  • grammar: good

Author:  Francesco Navarria  on 2022-02-11  [id 2193]

(in reply to Report 1 on 2022-02-02)
Category:
answer to question

Thank you for reading carefully the manuscript and for your comments. I have tried to address all your points.

  1. Fig. 3 had a rather poor contrast: I modified it so that the scales with the number of events are now clearly visible, and the labels are easier to read. I also referred to their difference in the text. The purpose of the figure is to show which part of the flux is accepted by the SND detector in the location chosen. One point which is underlined in the text is the correlation, due to the pT of the neutrino, between E and η for neutrinos coming from the pp intersect. This is clearly visible for ντ, where one can see the two lines corresponding to the two neutrinos from Ds decay. On the other hand, the νμ flux especially at low E is dominated by π and K decays.

  2. Figure 4 shows the number of neutrinos interacting via CC deep-inelastic scattering in SND, as it is said now in the text, so it is basically flux times target mass times cross section, with the cross sections extrapolated from the measured energy range for νμs, based on the Standard Model expectations. For ντ mass effects are included, which tend to reduce the cross section.

  3. I am not sure to understand your point. Our aim is to use observed νe events, assumed to have interacted with a SM cross section, to derive information on charm production at high eta. The quoted 35% systematic uncertainty is estimated from the kaon νe subtraction (20%), the pp → νeX cross section determination (basically the unfolding procedure, 20%) and the correlation between the charmed hadron yield in a given eta region with the neutrinos in the measured eta region (based on a full simulation, 25%). As far as I can understand the extrapolation from a lower η interval and the related theory uncertainty does not play any role. However, as you say, this large uncertainty, which translates into the determination of the neutrino flux, is what convinced us to go rather for a ‘collider’-type measurement, than for a 'fixed target'-type cross section measurement.

  4. Your remark is correct. The text has been modified accordingly.

A new version will soon be uploaded to arXiv.

Anonymous on 2022-02-14  [id 2198]

(in reply to Francesco Navarria on 2022-02-11 [id 2193])

I thank the author for making the clarifications, and responding to all of my comments.

For point (3), I had incorrectly understood that the systematic uncertainty quoted for the charm hadron yield in Table 1 was for the absolute rate (i.e. a theory systematic on pp -> (D+X)-> nu + X’). I understand now that the quoted systematic yield uncertainty (from theory) is the modelling uncertainty between the neutrino and D-hadron level cross section (at a differential level). If it is known what that uncertainty is dominated by (PDF knowledge, missing higher order effects), i.e. that which gives the dominant part of the 25% systematic, stating that could be useful.

Anonymous on 2022-02-16  [id 2212]

(in reply to Anonymous Comment on 2022-02-14 [id 2198])
Category:
answer to question

I don’t think that there is an easy/quick way of adding a few words. The uncertainty due to varying the PDFs turns to be important when the eta of the charmed hadron is lower than that of the neutrino in the SND acceptance (cases with equal, higher or lower eta were analyzed separately). However, it isn’t the dominant uncertainty, that due to hadronisation is larger in all cases, and at low charmed hadron eta that from the variation of the factorization and renormalization scales dominates. Varying the mass of the charmed quark also plays a role. Mentioning all the contributions would perhaps imply describing too many details of the analysis. Even if there are significant changes in the total rates of charm production and in the subsequent hadronisation, results of varying the simulations show that the migration in pseudo-rapidity between charmed hadrons and electron neutrinos is stable within about 25%.

Subir Sarkar  on 2022-02-17  [id 2213]

(in reply to Anonymous Comment on 2022-02-16 [id 2212])

Dear author, thank you for the clarification. I agree this is better discussed in a full paper so I am approving publication of this submission for a Proceedings.

Thanks also to the Referee for bringing up some important issues.

---

## Editorial Decision

resubmitted